# THE RIGHT TO BE FORGOTTEN IN PRUNING: UNVEIL MACHINE UNLEARNING ON SPARSE MODELS

## ABSTRACT

Machine unlearning aims to efficiently eliminate the memory about deleted data from trained models and address the right to be forgotten. Despite the success of existing unlearning algorithms, unlearning in sparse models has not yet been well studied. In this paper, we empirically find that the deleted data has an impact on the pruned topology in a sparse model. Motivated by the observation and the right to be forgotten, we define a new terminology "un-pruning" to eliminate the impact of deleted data on model pruning. Then we propose an un-pruning algorithm to approximate the pruned topology driven by retained data. We remark that any existing unlearning algorithm can be integrated with the proposed un-pruning workflow and the error of un-pruning is upper-bounded in theory. Also, our un-pruning algorithm can be applied to both structured sparse models and unstructured sparse models. In the experiment, we further find that Membership Inference Attack (MIA) accuracy is unreliable for assessing whether a model has forgotten deleted data, as a small change in the amount of deleted data can produce arbitrary MIA results. Accordingly, we devise new performance metrics for sparse models to evaluate the success of un-pruning. Lastly, we conduct extensive experiments to verify the efficacy of un-pruning with various pruning methods and unlearning algorithms. Our code is released at `https://anonymous.4open.science/r/UnlearningSparseModels-FBC5/`.

## 1 INTRODUCTION

The *Right to be Forgotten* is an emerging principle in artificial intelligence (AI) outlined by regulations such as the General Data Protect Regulation (GDPR), the California Consumer Privacy Act (CCPA), and Canada's proposed Consumer Privacy Protection Act (CPPA) Biega & Finck (2021); Regulation (2016); OAG (2021). A straightforward "forgetting" strategy is to retrain new models from scratch on the remaining data as if the deleted data has never been seen by the model. However, the retraining method is impractical due to the expensive cost of frequent deletion requests over complex models. To resolve this challenge, a plethora of machine unlearning methods have been developed to efficiently update the model without retraining Thudi et al. (2022); Schelter (2019); Bourtoule et al. (2021); Mahadevan & Mathioudakis (2021b); Zhang et al. (2024b); Liu et al. (2024a;b); Golatkar et al. (2020b); Xu et al. (2023); Pal et al. (2025); Liu et al. (2025); Fan et al. (2025); Sun et al. (2024); Zhuang et al. (2024); Fan et al. (2024); Zhou et al. (2025); Khalil et al. (2025); Spartalis et al. (2025).

Despite the success of existing unlearning algorithms, unlearning on sparse models has not been extensively studied yet. We remark that unlearning on sparse models is nontrivial. In Figure 1(a), we visualize the difference between a sparse model based on the original data and a retrained model based on the remaining data (5% data deletion). Specifically, we leverage the Lottery Ticket Hypothesis (LTH) Frankle & Carbin (2019) to prune a model with the original training data and the remaining data respectively. We can observe that most indexes of the pruned parameters are different between two sparse models. There is only around a 58% overlap of pruned indexes across two sparse models (60% sparsity) of a ResNet-18. The observation also holds for structured pruning. As shown in Figure 1(b), there are significant differences regarding pruned channels. The results indicate that the pruned indexes are data-dependent.

Table 1: Terminologies.

| Terminology | Data used | Input model | Description |
|---|---|---|---|
| Retraining | Retained data | Dense Model | Train the model from scratch |
| Unlearning | Retained and/or deleted data | Dense Model | Generate an unlearned model as though it has never seen the deleted data |
| Pruning | Full dataset | Dense Model | Prune a dense model to a sparse model with a sparse topology |
| Retraining + repruning | Retained dataset | Dense Model | Train and prune the model from scratch |
| Un-pruning | Retained and/or deleted data | Sparse Model | Update the topology of the sparse model to align with retraining + repruning |

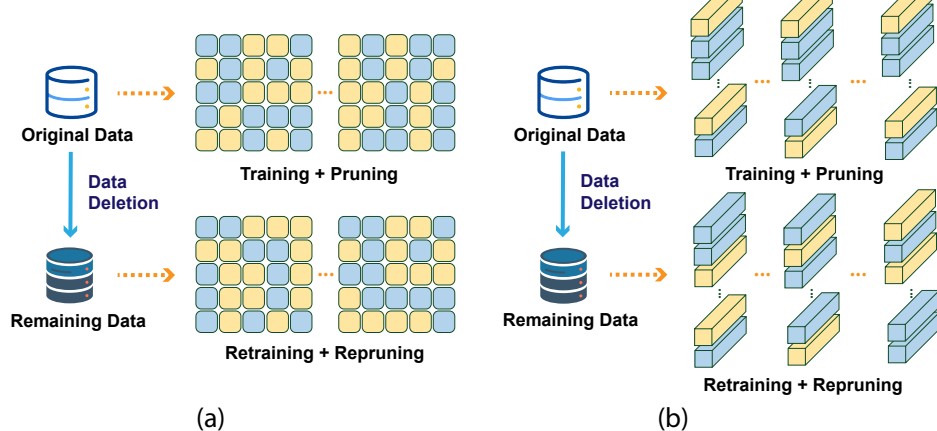

Figure 1: (a) Pruning with the original data vs. repruning with the remaining data. The blue blocks indicate pruned parameters. (b) Structured pruning with the original data vs. the remaining data. The blue blocks indicate pruned channels.

Motivated by this observation, we raise a new research problem: **how to eliminate the influence of deleted data on the pruned topology in a sparse model?** More specifically, the problem is how to generate a new sparse model

Table 2: Reconstruction with pruned models

| Method | Del. Ratio | Extraction-Score($\downarrow$) | SSIM($\uparrow$) |
|---|---|---|---|
| Original | 0 | 5295.46 | 0.4438 |
| Un-prune | 20% | 5786.37 | 0.3787 |
| Un-prune | 40% | 5841.08 | 0.3521 |

as if the model has never seen the forgotten data. We define "un-pruning" as a new terminology for eliminating the impact of deleted data on pruning. Table 1 lists the difference between un-pruning and other terminologies such as unlearning. We highlight that un-pruning is non-trivial because: (1) while retraining + repruning serves as the gold standard, it is impractical to perform retraining + repruning on a large-scale model for frequent data deletions. (2) existing unlearning algorithms on the remaining parameters in a sparse model cannot change the pruned topology. These algorithms based on approximating activation values may only align the model's accuracy with the retrained model but fail to align the topology with the standard retraining + repruning Wang et al. (2024); Qu et al. (2023); (3) users have the right to withdraw the influence of their data on pruning processes. Our empirical results demonstrate that deleted data has a significant impact on the topology of the sparse model. Consequently, the continued use of sparse models derived from deleted data instances can be deemed illegal Fosch-Villaronga et al. (2018). (4) pruning has been considered as a process of learning from the training data Zhou et al. (2019a); Tung & Mori (2018); Haim et al. (2022b). Correspondingly, un-pruning can be considered as a process of unlearning. Thus, updating the topology of a sparse model provides us with a new way of unlearning. We use Haim et al. (2022a) to reconstruct training data with the sparse model. Compared with the original sparse model (ResNet-18, 40% sparsity), the un-pruned model is less accurate in reconstructing the training data.

In this paper, we propose an "un-pruning" algorithm that allows us to integrate any existing unlearning methods into our un-pruning process. Specifically, we activate the pruned parameters to enable topology change in the sparse model. Then we leverage existing unlearning algorithms to approximate the new parameters and new topology by estimating the influence of deleted data on learning and pruning. We grow the topology in an interactive way to allow smooth and steady topol-

ogy change. Note that the updated model has to be pruned to the original sparsity so that it can be compared with the retaining + repruning strategy. We highlight that our algorithms can be applied to both unstructured and structured topology. We also analyze our un-pruning algorithm in theory. Compared with retaining + repruning, the un-pruning error regarding the mask is upper bounded and characterized by the model sparsity.

To verify the effectiveness of un-pruning, we follow existing unlearning works to report TA (test accuracy), UA (unlearning accuracy), and MIA (Membership Inference Attack). However, we empirically find that MIA is not a reliable metric to measure the quality of unlearning. With a small change in the data proportion of the binary classification model (i.e., training data vs. test data), the MIA value can fall anywhere within the range of 0 to 1 randomly. Research in large language models also found that the results of MIA are almost equivalent to guesswork Duan et al. (2024); Liu et al. (2025); Das et al. (2024); Zhang et al. (2024c). In this paper, we devise new performance metrics to evaluate the quality of unlearning. Specifically, we calculate the structural similarity between neural network representations Klabunde et al. (2023); Kornblith et al. (2019); Xu et al. (2024) to measure the unlearning quality. Since retraining + repruning serves as the golden standard, we calculate the intersection of pruned indexes between a repruned sparse model based on remaining data and an "un-pruned" sparse model at the same sparsity level. Intuitively, a high intersection ratio means a high similarity between two topologies. In the experiment, we verify the effectiveness and efficiency of our algorithms with various unlearning algorithms and various unlearning algorithms. The contribution of this work can be summarized as:

- We conduct empirical studies to investigate the data-dependency of pruning.
- We raise a new research problem: how to eliminate the influence of deleted data on the pruned topology of a sparse model?
- We propose an un-pruning algorithm to approximate the pruned topology driven by the retained data without costly retraining and repruning. Our algorithm can integrate any unlearning algorithm into both unstructured topology and structured topology.
- We design new performance metrics and conduct extensive experiments to verify the effectiveness and efficiency of our un-pruning algorithm.

## 2 PRELIMINARY

**Machine Unlearning.** Given a model $A$ trained on a complete dataset $D$, the user can request to remove their data $D_f$ from $D$. Machine unlearning aims to eliminate the influence of $D_f$ on $A$ and unlearn a new model that forgets what has been learned from $D_f$. Denote $D_r$ as the remaining data where $D = D_f \cup D_r$. A naive solution is to retrain a new model $A_r$ based on $D_r$ Dr) from scratch. However, this solution is impractical due to the high computational cost for frequent data deletions over complex models. Recently, approximate unlearning algorithms Zhang et al. (2024b); Liu et al. (2024a;b); Park et al. (2024a); Chien et al. (2024) have been developed to directly generate a new model $A_u$ from $A$ without retraining such that the unlearned model $A_u$ approximates $A_r$ as though it had never seen $D_r$: $A_r(\cdot) \approx A_u(\cdot)$.

**Pruning.** Pruning can be categorized into two types: unstructured and structured pruning Cheng et al. (2024a); He & Xiao (2023); Xu et al. (2020); Cheng et al. (2024b); Tang et al. (2024); Ma & Niu (2018); Deng et al. (2020). Denote $\Theta$ as the parameters of a model $A$. Unstructured pruning aims to associate a binary mask $M$ where $|M| = |\Theta|$ and a zero value in $M$ indicates that the corresponding parameter is masked out by $\Theta \odot M$. For example, the Lottery Ticket Hypothesis (LTH) Frankle & Carbin (2019) masks out the parameters with the smallest magnitude. Instead of pruning parameters distributed irregularly on the model framework, structured pruning removes entire filters, channels, or even layers. Given a specific prune ratio and a neural network with $A = \{s_1, s_2, \cdots, s_L\}$, where $s_i$ can be the set of channels, filters, neurons, or dense layers, the target is to search for $A' = \{s'_1, s'_2, \cdots, s'_L\}$ to minimize performance degeneration and maximize speed improvement under the given prune ratio, where $s'_i \subseteq s'_i$ and $i \in \{1, 2, \cdots, L\}$.

**Problem setup.** How to eliminate the influence of deleted data on pruning is an open problem. As evidenced by our empirical study in Figure 1, the pruned structure is data-dependent. The user has the right to withdraw the influence of their data on pruning as the continued use of sparsed models pruned based on deleted data instances can be deemed illegal Fosch-Villaronga et al. (2018). Let $A$ be the sparse model parametrized by $\Theta \odot M$ where $\Theta$ is the set of weights and $M$ is the mask. Suppose the sparse $A$ is trained and pruned based on the original dataset $D$. Given a sub-dataset

$D_f \subseteq D$ removed from $D$, the problem is to generate a new sparse model $A_u$ parametrized by $\Theta_u \odot M$ where $\Theta_u$ is the new parameters and $M_u$ is the new mask. Considering that the retraining+repruning strategy is impractical for frequent data deletions and expensive pruning strategies such as iterative pruning, our target is to directly generate $\Theta_u$ and $M_u$ without retraining and repruning such that $M_u$ is distinguishable from $M$ and $\Theta_u \odot M_u$ is indistinguishable from the result of retraining+repruning.

## 3  UN-PRUNING

The first challenge of this research problem is that the pruned parameters are frozen during the training process. Therefore, traditional unlearning algorithms that perform parameter updates can not change the value of frozen parameters. A prior work (Jia et al., 2023) has also studied unlearning on sparse models. Unfortunately, this work does not update the mask during the unlearning process (i.e., the right to be forgotten in pruning has not been addressed). In this paper, we propose an "un-pruning" algorithm to directly generate the new parameters and the new mask without retraining and repruning. Without loss of generality, Algorithm 1 demonstrates our un-pruning process.

---
**Algorithm 1** Un-pruning

**Input:** Model parameters $\Theta$, Mask $M$, original dataset $D$, deleted dataset $D_f$, original sparsity $s_\Theta\%$, un-pruning ratio $p_\Theta\%$, unlearning algorithm $U$, , un-pruning iterations $T$

**Output:** New parameters $\Theta_u$, new mask $M_u$

---
1: **for** each iteration t= in $0, 1, 2, \cdots, T-1$ **do**
2:     Randomly re-initialize pruned parameters as non-zero values: $\Theta \leftarrow \Theta + (1-M)\Theta_0$
3:     Perform unlearning (any unlearning algorithm) on the dense full model $\Theta \leftarrow U(\Theta, D_f, D)$
4:     Update mask $M$: set $p_\Theta\%$ mask indexes whose corresponding parameters have the highest magnitude values as 1;
5:     Update parameters: $\Theta \leftarrow \Theta \odot M$
6: **end for**
7: Prune the new model to the original sparsity $s_\Theta\%$ by masking the lowest magnitude values; set the corresponding index in $M$ as 0.
8: $\Theta_u = \Theta, M_u = M$

---

In each iteration, we first unfreeze and re-initialize the pruned parameters to perform unlearning algorithms. We re-initialize these parameters because we find that zero-value parameters will remain unchanged with most unlearning algorithms and the pruned indexes will not change after we perform unlearning. In this paper, we have introduced two initialization strategies: original initialization and random initialization. We have investigated the difference in the experiment. Then we update the sparse model by rolling back $p_\Theta\%$ pruned indexes whose corresponding parameters have the highest magnitude values, i.e., the sparsity will be $(s_\Theta - p_\Theta)\%$ after the first iteration. Note that we will update the mask and parameters after unlearning in each iteration. After $T$ iterations, the sparsity will be $(s_\Theta - T * p_\Theta)\%$. Note that we prune the model to the original sparsity $s_\Theta\%$ with one-shot pruning for a fair comparison with repruning.

In this paper, we have investigated 6 most popular approximate unlearning methods: GradientAscent Graves et al. (2021), SCRUB Kurmanji et al. (2023), Fisher Golatkar et al. (2020a); Foster et al. (2024b); Kirkpatrick et al. (2017), WoodFisher Singh & Alistarh (2020), CertifiedUnlearning Zhang et al. (2024a), SFRON Huang et al. (2024). While these unlearning methods vary on the objective function, all existing unlearning algorithms can be integrated with our method.

We remark that our un-pruning strategy works for these unstructured turning algorithms that do not require extra parameters for pruning. Similarly, in each iteration, we unfreeze the most important channels, filters, or dense layers. Specifically, we use the same standard of pruning structures to choose important structures. For example, Soft Filter Pruning He et al. (2018) selects filters to be pruned based on the $l_2$ norm of filters in the training stage. Accordingly, we can select the filters to recover based on the $l_2$ norm. Note that there are some structured pruning algorithms with extra parameters that are used to learn which part to pruneBiP Zhang et al. (2022b); Sehwag et al. (2020); Wang & Fu (2022). For those algorithms with extra learnable parameters, we advocate for new algorithms to address the extra parameters since existing algorithms only update the model.

**Evaluation of un-pruning.** While the similarity between the unlearned model and the retrained model is usually used to measure the quality of unlearning algorithms Xu et al. (2024), it is still an open problem to measure the structural similarity between the "un-pruned" model and the standard

"repruned" model. First, we define the intersection ratio and the union ratio of two pruning models:

$$IoM = \frac{\|M_u \odot M_r\|_1}{N} \tag{1}$$

$$UoM = \frac{\|M_u + M_r - M_u \odot M_r\|_1}{N}, \tag{2}$$

where $\|\cdot\|_1$ is the number of 1 in the mask and N is the total number of parameters. We use $M_u$ and $M_r$ to represent the un-pruned mask and re-pruned mask. Based on the intersection and the union, we can further define IoU (intersection of union):

$$IoU = \frac{\|M_u \odot M_r\|_1}{\|M_u + M_r - M_u \odot M_r\|_1} \tag{3}$$

Figure 2 visualizes the intersection ratio between "un-pruning" integrated with different unlearning algorithms. We also show the unlearning accuracy and the results indicate that our un-pruning algorithm can also guarantee the unlearning accuracy.

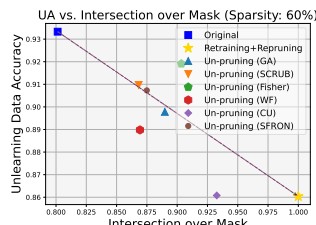

Besides the structural similarity, we have also introduced the KL-divergence Golatkar et al. (2020a) to measure the difference:

$$K(\Theta_u, \Theta_r, M_u, M_r) = \text{KL}(P(M_{u,i} \odot \Theta_{u,i}) \| P(M_{r,i} \odot \Theta_{r,i})) \tag{4}$$

where $P(\cdot)$ is the distribution probability introduced by the randomness in the learning algorithm. Intuitively, the unlearning process will provide an exact unlearning if $K(\theta_u, \theta_r) = 0$.

Figure 2: Intersection ratio and unlearning accuracy.

**Analysis of un-pruning.** Given weights $\Theta$ and pruning mask $M$, the pruning target can be simplified as:

$$\min L(\Theta; M; D) = \min \frac{1}{|D|} \sum_{i=1}^{|D|} l(\Theta \odot M; x_i, y_i), \tag{5}$$

where $(x_i, y_i)$ is a data point in $D$ and $l$ is the loss function.

Given $k$ components in the model: $\Theta = \Theta_0, \Theta_1, ..., \Theta_k$, we follow Unrolling SGD: Thudi et al. (2022) to write the update the $j$-th parameter of $i$-th component in pruning iteration $s$ in the unlearning process as:

$$\Theta_{i,j}^s = M_{i,j}^{s-1}(\Theta_{i,j}^{s-1} - \eta \nabla_\Theta l(\Theta \odot M; D_f), \tag{6}$$

where $\eta$ represents the unlearning rate. To establish a connection between un-pruning and unlearning, we rewrite the pruning strategy as:

$$M = g(\Theta) = Sigmoid(\frac{\log P(\Theta) + \epsilon}{\tau}), \epsilon \sim Gumbel(0, 1) \tag{7}$$

where probability $p$ represents the sampling probability of the parameters over the random learning algorithm.

Start with $\tilde{\Theta}^0 = \Theta^S$ after $S$ iterations of training and pruning, we roll back the parameters by:

$$\begin{aligned}
\tilde{\Theta}^t &= g(\tilde{\Theta}^{t-1})(\tilde{\Theta}^{t-1} + \eta \frac{\partial L((g(\tilde{\Theta}^{t-1})\tilde{\Theta}^{t-1}; D_f)}{\tilde{\Theta}^{t-1}}) \\
&= g(\tilde{\Theta}^{t-1})(\tilde{\Theta}^{t-1} + \eta \frac{\partial \frac{1}{|D|} \sum_{i=1}^{|D|} l(\tilde{\Theta} \odot M; x_i, y_i)}{\tilde{\Theta}^{t-1}})
\end{aligned} \tag{8}$$

Since the new mask $M_t$ is data-dependent, the final mask of the given parameter is:

$$M = g(\tilde{\Theta}^t)g(\tilde{\Theta}^{t-1}) \ldots g(\tilde{\Theta}^1)g(\tilde{\Theta}^0) = \prod_{i=0}^{t} g(\tilde{\Theta}^i) \tag{9}$$

To estimate the un-pruning error regarding $M$, we approximate the unlearning error in Jia et al. (2023) in the un-pruning process. Specifically, we first approximate the loss function using the second-order Taylor expansion around the initial mask $M^0 = M$:

$$L(M) = L(\mathbf{1}) + \frac{1}{2}(\mathbf{1} - M)^T \mathbb{E}[\frac{\partial^2 L(\mathbf{1})}{\partial M^2}](\mathbf{1} - M) \tag{10}$$

Denote the Hessian matrix of the loss function with:

$$H = \nabla^2 L(\tilde{\Theta}; D_f) \tag{11}$$

At this moment, the changes in parameters are:

$$\Delta(\tilde{\Theta}) := \tilde{\Theta} - \tilde{\Theta}^0 \approx H^{-1} \nabla L(\tilde{\Theta}_o; D_f) \tag{12}$$

Then let the $\sigma$ be the largest singular value:

$$\lambda(M) := max\{\lambda_j(H), 1\} \tag{13}$$

According to Thudi et al. (2022), the overall parameters changes can be written as:

$$\tilde{\Theta}^t \approx \tilde{\Theta}_0 + \eta \sum_{i=0}^{t-1} \frac{\partial L}{\partial \tilde{\Theta}} + \sum_{i=1}^{t-1} k(i) \tag{14}$$

where $k(i)$ is defined recursively as:

$$k(i) = -\eta \frac{\partial^2 L}{\partial^2 \tilde{\Theta}}(k(i-1)), \; k(0) = 0 \tag{15}$$

By introducing the mask, the parameter change can be rewritten as:

$$\tilde{\Theta}^t \approx \tilde{\Theta}_0 + \eta M^{t-1} \sum_{i=1}^{t-1} \nabla L(M^{t-1} \odot \tilde{\Theta}; D_f) + M^0 \sum_{i=1}^{t-1} k(i) \tag{16}$$

By comparing Eq. (6) and Eq. (16), we can see that the un-pruning error between $\tilde{\Theta}^t$ and the original $\Theta_0$ is:

$$e(g(\Theta)) = \left\| M^0 \sum_{i=1}^{t-1} k(i) \right\|_2 \approx \eta^2 \left\| \text{diag}(^0) \sum_{i=1}^{t-1} \nabla^2 l(\Theta_0, D_f) \sum_{j=0}^{i-1} M^0 \odot \nabla l(\Theta_0, D_f) \right\|_2 \tag{17}$$

According to the Triangle Inequality:

$$e(g(\Theta)) \leq \frac{\eta^2}{2}(t-1)\|M^0 \odot (\Theta^t - \Theta_0)\|_2 \lambda(M^0) \tag{18}$$

In summary, the upper bound of the un-pruning error regarding the mask can be characterized by the largest singular value $\lambda(M)$ and the model sparsity:

$$e(g(\Theta)) = O(\eta^2 t \|M \odot \Theta\|_2 \lambda(M)) \tag{19}$$

The theory can be empirically justified by Table 3. As the model sparsity increases, the KL divergence between "un-pruning" and "repruning" also increases. It verifies that the un-pruning error bound is related to the sparsity.

Table 3: KL divergence at different sparsity levels.

| Method | KL | | |
|---|---|---|---|
| | 40% | 60% | 95% |
| Original | 19.06 | 19.68 | 21.79 |
| Retrain | 0.00 | 0.00 | 0.00 |
| Fisher | 16.47 | 16.85 | 15.09 |
| WoodFisher | 19.06 | 19.68 | 21.79 |
| GradientAscent | 19.05 | 19.66 | 21.79 |
| CertifiedUnlearning | 16.49 | 19.32 | 19.08 |
| SCRUB | 19.04 | 19.57 | 21.54 |
| SFRon | 19.03 | 19.60 | 21.59 |

To further investigate the KL divergence after un-pruning, we measure the difference between the repruned model and retrained model, we follow (Dziugaite & Roy, 2017) to introduce the inequality:

$$\mathbb{E}_\Theta L(\Theta; M; D) \leq \mathbb{E}_\Theta L(\Theta; M; D_r)$$
$$+ \sqrt{\frac{\text{KL}(Q\|P) + \log \frac{|D_f|}{\delta}}{2(|D_r| - 1)}}, \forall \delta > 0, \tag{20}$$

where $Q$ and $P$ are prior and posterior distribution on $\Theta$.

$$\text{KL}(Q||P) \leq \frac{1}{2}\sum_i(1 + k_i log(\frac{1 + \alpha_1^2||\Theta_i^t - \Theta_i^0||}{k_i\sigma_{Q,i}^2})), \quad (21)$$

where $k_i$ is the number of parameters in module $i$ of $\Theta^t$ and $\sigma$ is the standard deviation of Gaussian noise. Then we have the following inequality Chatterji et al. (2020):

$$\mathbb{E}_\Theta L(\Theta; M; D) - \mathbb{E}_\Theta L(\Theta; M; D_r)$$
$$\leq \sqrt{\frac{\frac{1}{4}\sum_{i=1}k_i\log(1 + \frac{\mu_{i,\epsilon}(\Theta)}{n_i}) + \log(\frac{|D_r|}{\delta}) + \tilde{O}(1)}{|D_r| - 1}}, \quad (22)$$

where

$$\mu_{i,\epsilon}(\Theta) = \min_{0 \leq \alpha_i, \sigma_i \leq 1}\{\frac{\alpha_i^2||\Theta^t - \Theta_0^i||^2}{\sigma_i} :$$
$$\mathbb{E}_{u \sim N(0,\sigma_i^2)}L(\theta_i; D_r) \leq \epsilon\} \quad (23)$$

and $\alpha_i$ is the re-initialized weights. In inclusion, the difference between un-pruning and repruning is bounded.

## 4 EXPERIMENT

In this paper, we have followed (Jia et al., 2023) to set up the experiment. Specifically, we have introduced 3 models to be pruned: **ResNet-18** and **AlexNet**, and **ViT**. The models are trained and pruned on 3 datasets: **CIFAR-10**, **FashionMNIST**, and **ImageNet**. For all the datasets and model architectures, we randomly deleted 10% data. We use (LTH) Frankle & Carbin (2019) and Soft Filter Pruning He et al. (2018) as the default unstructured pruning and structured pruning method respectively. The sparse model can reconstruct training data, suggesting that their reconstruction ability is linked to data memorization Haim et al. (2022b). Inspired by this, we evaluate whether a sparse ResNet-18 model (with 40% sparsity) retains training data by testing its reconstruction ability. Besides the intersection between the repruned mask and the "un-pruned" mask, we have also introduced IoM(Intersection over Mask), TA (test accuracy), UA (unlearning accuracy), and MIA. We highlight that MIA is fragile as an evaluation of unlearning. We will demonstrate our findings in Section 5.

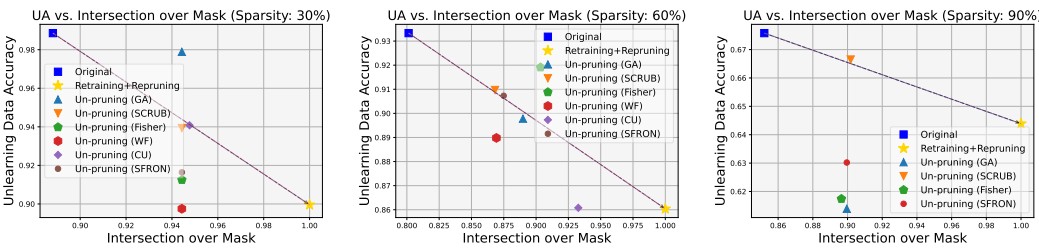

Figure 3: Structured un-pruning (IoM).

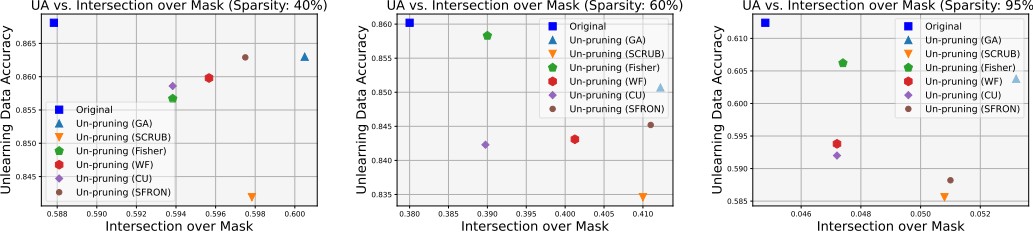

Figure 4: Unstructured un-pruning (IoM).

**Results and analysis.** For Table 2, Extraction-Score refers to the distance between generated images and real images. SSIM refers to the structural similarity between generated images and real images. The experiments shows that the new (un-pruned) sparse model is less accurate in reconstructing the

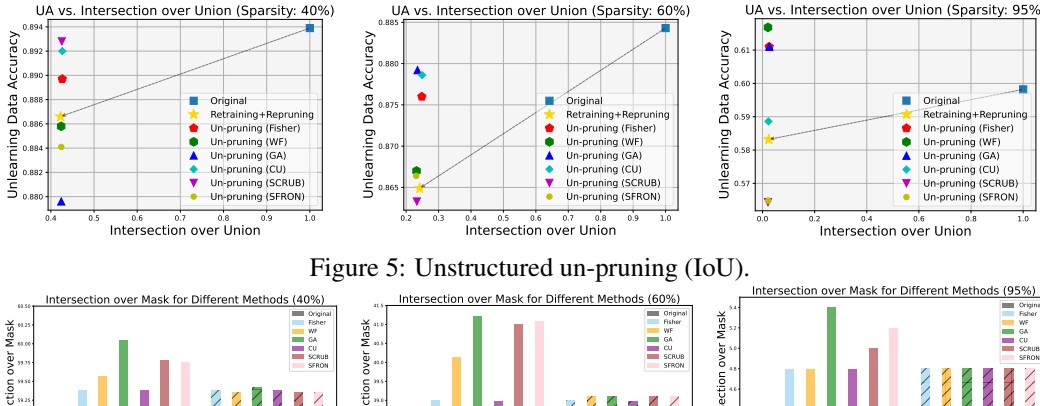

Figure 5: Unstructured un-pruning (IoU).

Figure 6: Original initialization vs. random initialization.

training data compared with the original sparse model. We integrated various unlearning algorithms including Fisher Golatkar et al. (2020a), WoodFisher (WF) Singh & Alistarh (2020), Gradient Ascent (GA) Graves et al. (2021), Certified Unlearning (CU) Zhang et al. (2024a), SCRUB Kurmanji et al. (2023), and SFRON Huang et al. (2024) into our un-pruning method. Figure 3, 4 and 5 show the experiment result for 3 sparsity levels (i.e., 40%, 60%, and 95%) with structured pruning and structured pruning, respectively. The x-axis indicates IOU and the y-axis measures UA. Figure 7 visualizes the overall performance including sparsity, TA, and UA. Compared with the original model, the un-pruned model is closer to the retrained + repruned model in terms of both IoM and UA. At the same time, the un-pruned model has a very different topology and UA from the original model. Also, we find that our un-pruning has a better performance on structured pruning. This is because structured topology is easier to approximate than unstructured topology where there is a marginal difference in the parameter's magnitude within a component (e.g., channel). The results verify that our un-pruning algorithm can generate a new sparse model that is similar to retraining + repruning.

**Original initialization vs. random initialization.** We analyze experiments with two initialization methods. For the original initialization, we save the original initialization weights and restore the initialization weights in the un-pruning process. For the random initialization, we assign the random values to pruned parameters. As shown in Figure 6, the original initialization outperforms the random initialization. This is because the original initialization provides a more stable parameter

Table 4: Running time.

| Method | Running Time (s) | | |
|---|---|---|---|
| | 40% | 60% | 95% |
| Retraining+Repruning | 6749 | 12217 | 39235 |
| Fisher | 189 | 182 | 184 |
| WF | 50 | 48 | 48 |
| GA | 1793 | 1778 | 1779 |
| CU | 878 | 908 | 846 |
| SCRUB | 133 | 132 | 134 |
| SFRon | 91 | 89 | 88 |

distribution, thereby achieving better performance in the IoM metric. The result aligns with the viewpoint of Zhou et al. (2019b) that for a given initialized network, there exists a supermask that performs best on a specific dataset. The original initialization helps us approach the supertask for the remaining dataset, thereby facilitating the forgetting of the unlearning data.

**Running time analysis** In this part, we present and analyze the running time of experiments in Table 4. For an iterative pruning method, the running time of retraining increases as the sparsity increases. Compared with retraining + repruning, our un-pruning method can significantly reduce the computational complexity. At the same time, there is a difference in terms of the running time when we integrate different unlearning algorithms.

## 5 VULNERABILITY OF MIA IN MACHINE UNLEARNING

We initially planned to use MIA as an evaluation metric to calculate our baseline results. However, during the experiments, we found that MIA does not effectively verify the model's unlearning per-

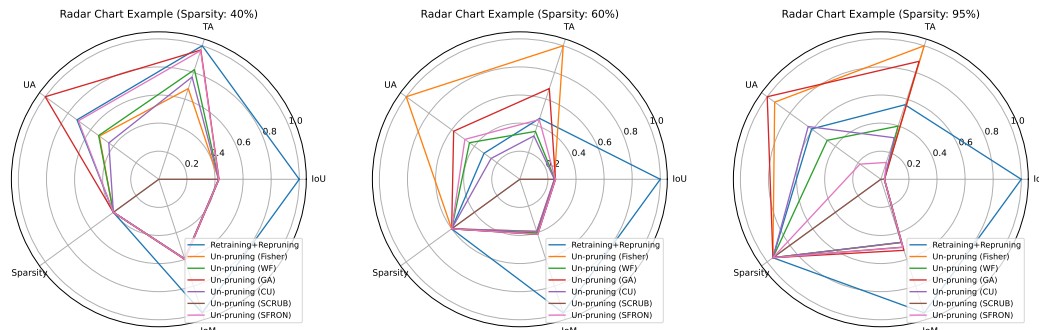

Figure 7: Radar charts of 5 performance metrics.

formance. We recorded five MIA evaluation indicators: MIA's Correctness, Confidence, Entropy, m-Entropy, and Probability. However, during the evaluation process of correctness, we found that MIA is close to random guessing, which means it is difficult to determine whether the data belongs to the training set. Then, by fine-tuning the ratio of the training set to the test set, we found that when the sample size exceeded a certain range, MIA would maintain its original value. However, when the sample size exceeded or fell below a certain threshold, MIA would rise to 1.0 or decrease to 0 (See in Figure 8). For the other four indicators, by fine-tuning the ratios, we can obtain values in any range from 0 to 1 (See in Figure 9 and appendix). At the same time, it should be noted that the ratio is always maintained at around 1.0, which means that only a few samples need to be reduced or increased to manipulate MIA. Therefore, using MIA's evaluation on unlearning is fragile, and it is even impossible to distinguish whether the retrained model has forgotten the deleted data.

## 6 DISCUSSION

As Figure 10 11 and 12 shown, MIA would be significantly changed by slightly adjusting a bit of samples. Therefore, even when the training and test sets in the shadow dataset are approximately in a 1:1 ratio, adjusting the data within the error margin (±5%) can yield completely opposite results. Due to the low robustness of the results, we consider MIA to be unreliable for evaluating unlearning in sparse models.

As Tab 7 mentioned, each row contains several sparsifications (i.e., dense and a few sparsity-level) and 8 situations (Original, retraining, and 6 unlearn methods). We ensure that all experimental conditions remain consistent, with the only difference being in the training data (the original experiment uses the full dataset, while the retrain experiment uses a training set where 20% of the data is removed based on random seed 42). In some cases, high sparsity leads to a catastrophic collapse in accuracy (e.g., 90% and 95% sparsity sometimes result in only 10% accuracy). Therefore, we omit these clearly broken results when presenting our results.

We still need to find or establish better metrics to evaluate whether some unlearn methods remove the impact of the deleted data or not. We should comprehensively consider the impact of data on all aspects such as the performance of downstream tasks or the topology of sparse models (weight distribution, connectivity patterns, and gradient dynamics). Furthermore, expanding experiments to a wider range of model architectures and datasets would enhance the generalizability of findings. While our study focuses on a specific setting, evaluating unlearned methods in architectures like Transformers or DDPM could reveal model-dependent behaviors.

## 7 CONCLUSION

In this paper, we empirically find that pruning is data-dependent. To address the right to be forgotten in pruning, we propose an un-pruning algorithm to approximate the pruned topology driven by the retained data without costly retraining and repruning. We prove the difference between un-pruning and repruning is bounded in theory. The experiment verifies the effectiveness of our method.

## 8 ETHICS STATEMENT

This work does not involve human subjects, personally identifiable data, or sensitive information. We believe it does not raise ethical concerns.

## 9 REPRODUCIBILITY STATEMENT

We have made our code available in the anonymous project. Detailed hyperparameters, training procedures, and dataset processing steps are provided in the code file. Random seeds are fixed for reproducibility, and all results have been verified across multiple independent runs.

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

## A APPENDIX

### A.1 LLM USAGE STATEMENT

Large language models (LLMs) were employed to assist in refining the clarity, grammar, and style of the manuscript. The use of LLMs was limited to language polishing and did not contribute to the generation of substantive ideas, analyses, or results. All scientific content, interpretations, and conclusions are the authors' own.

### A.2 RELATED WORK

**Pruning.** As the size of deep learning models increases, model pruning has attracted attention to reduce the parameter and improve computational efficiency without compromising accuracy (LeCun et al., 1989; Liang et al., 2021; Yin et al., 2023a; Kumar et al., 2024; Wang et al.; Bai et al., 2024). Existing pruning technologies can be classified into two types: structured pruning and unstructured pruning. In general, structured pruning (He et al., 2017; Liu et al., 2017; Li et al., 2017; Hu et al., 2016; Wen et al., 2016; Hong et al., 2018; Nonnenmacher et al., 2022; Halabi et al., 2022; Yin et al., 2023b; Nova et al., 2023) remove sparse patterns such as layers and channels from the full model. Unstructured pruning (Han et al., 2015; Liu et al., 2019; Cao et al., 2019; Ren et al.,

2019; Zhang et al., 2018; Alizadeh et al., 2022; Liao et al., 2023) removes irregularly distributed parameters instead of entire weight structures (e.g., channels). Recent study shows that there is a dependence between pruned and the training data. Specific masks are more sensitive to the training data Zhou et al. (2019b). Liu et al. (2024a) proves that model Sparsitization can simplify the machine unlearning. Experiments by Zhang et al. (2022a) conclude that the influence of deleted data on the connections between layers of the model remains unchanged.

**Machine unlearning.** Machine unlearning aims to remove the impact of deleted on trained models. While the retraining strategy is intuitive and costly, exact unlearning aims to learn a model with the same performance as the retrained model Bourtoule et al. (2021); Huang et al. (2024); Yao et al. (2023); Zhao et al. (2024); Li et al. (2021); Mahadevan & Mathioudakis (2021a); Brophy & Lowd (2021); Ginart et al. (2019); Cao & Yang (2015); Ullah et al. (2021); Chen et al. (2021). Approximate unlearning methods update the model by approximating the impact the deleted data on the training model Chien et al.; Park et al. (2024b); Graves et al. (2021); Marchant et al. (2022); Wu et al. (2020); Neel et al. (2021); Ginart et al. (2019); Foster et al. (2024a); Chundawat et al. (2023). Despite the success of these unlearning algorithms on dense models, unlearning on sparse models has not been extensively investigated. To the best the author's knowledge, (Jia et al., 2023) has also studied unlearning on sparse models. However, this work does not update the mask during the unlearning process (i.e., the right to be forgotten in pruning has not been addressed).

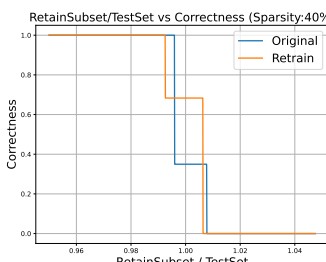 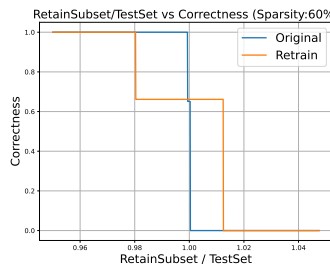 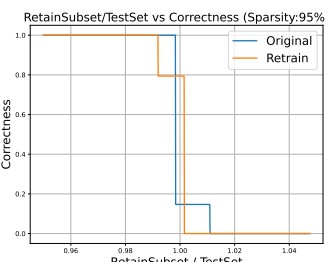

Figure 8: MIA correctness.

Table 5: Performance comparison on the ImageNet dataset (ResNet18)

| Method | IOU | IOM | TA | UA | Sparsity | Del. Ratio |
|---|---|---|---|---|---|---|
| Retraining + repruning | – | – | 56.57% | 54.87% | 19.03% | 20% |
| Un-pruning (WF) | 84.22% | 85.06% | 57.42% | 55.42% | 19.03% | 20% |

Table 6: Performance on ViT

| Method | IOU | IOM | TA | UA | Sparsity | Del. Ratio |
|---|---|---|---|---|---|---|
| Retraining + repruning | – | – | 72.68% | 85.29% | 41% | 20% |
| Un-pruning (WF) | 45.78% | 64.05% | 70.32% | 86.44% | 41% | 20% |

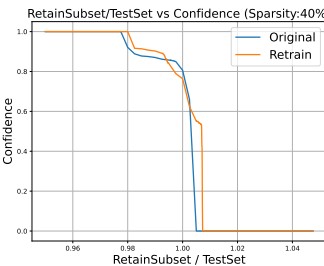 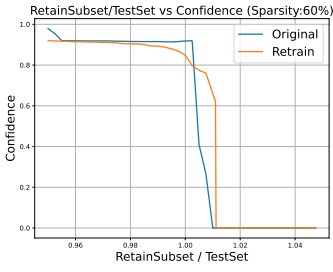 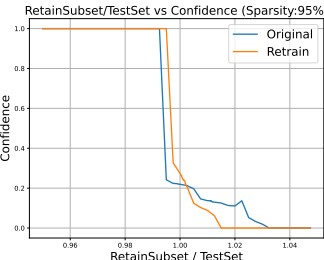

Figure 9: MIA confidence.

Table 7: Hyper Parameters Settings

| Type | Method | Model | Dataset | Learning Rate | Random Seed | Deleted Ratio |
|------|--------|-------|---------|---------------|-------------|---------------|
| Structured | He et al. (2018) | ResNet-20 | Cifar10 | 1e-3 | 42 | 20% |
| Unstructured | Frankle & Carbin (2018) | ResNet-18 | Cifar10 | 1e-3 | 42 | 20% |
| Unstructured | Frankle & Carbin (2018) | ResNet-18 | FashionMNIST | 1e-3 | 42 | 20% |

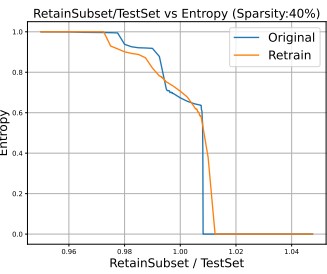 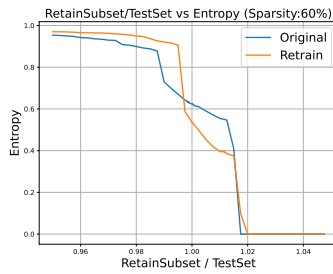 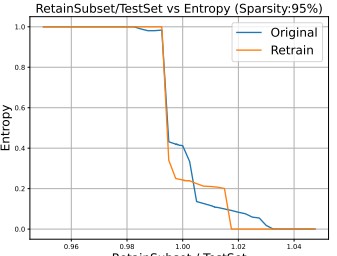

Figure 10: MIA Entropy.

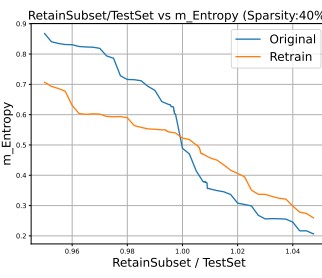 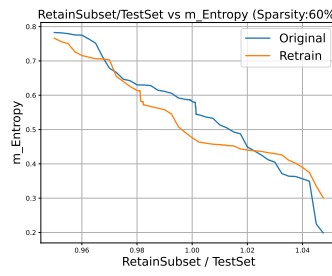 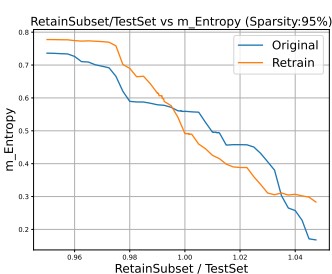

Figure 11: MIA m Entropy.

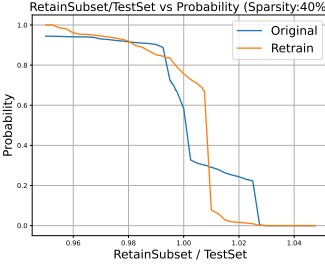 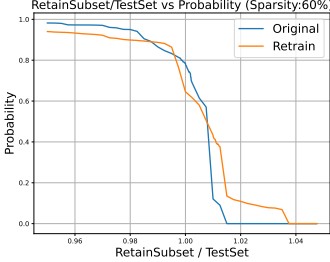 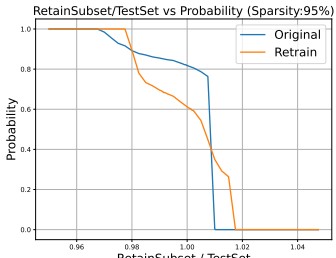

Figure 12: MIA Probability.

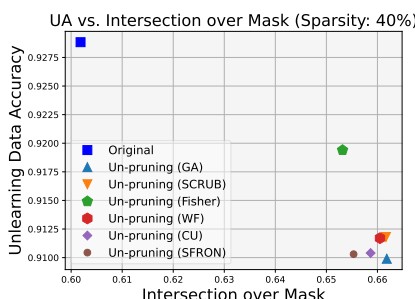 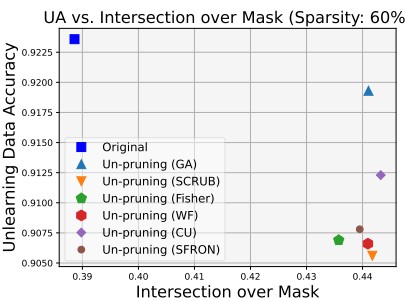

Figure 13: Unstructured un-pruning (IoM) in FashionMNIST

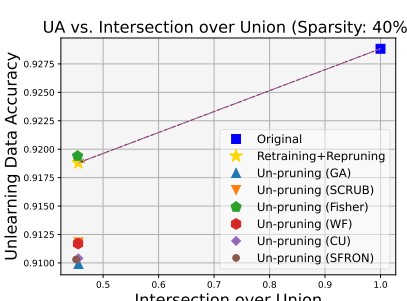 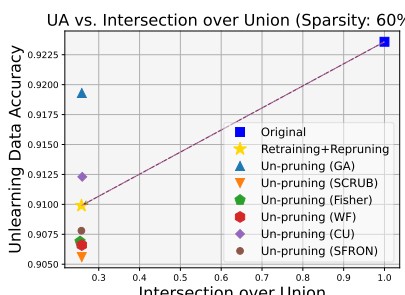

Figure 14: Unstructured un-pruning (IoU) in FashionMNIST.

