# OpenReview forum: "The Right to be Forgotten in Pruning: Unveil Machine Unlearning on Sparse Models"
_ICLR.cc/2026/Conference — ICLR 2026 Conference Withdrawn Submission_

### Official Review · Reviewer_Z9fi · 2025-10-27

**Soundness:** 2
**Presentation:** 1
**Contribution:** 2
**Rating:** 2
**Confidence:** 4

**Summary:**

This paper advances a study on *unpruning*, which analyzes model unlearning under a varying sparse topology. In addition, they question the effectiveness of MIA as a metric to verify unlearning and propose a different quantification metric.

**Strengths:**

- The setup of unpruning is interesting.
- The overall problem has potential to be relevant.

**Weaknesses:**

**Conceptual clarity and motivation:**
- The term *un-pruning* is a bit incorrect and imprecise. I understand the authors' idea of unifying unlearning and pruning, but un-pruning truly deviates from what the paper is presenting, and I do not think reflects “semantically” what is done in this paper. This could be more simply described as un-learning under a dynamic sparsity constraint. For what concerns naming, in addition, I do not honestly understand the point of using terms retraining and repruning for simply training and pruning: this is only adding more confusion.
- The authors claim in the introduction that un-pruning is non-trivial compared to un-learning, but then fail in providing a solid justification on why that is the case. Specifically, starting from line#090: (1) is true also for normal unlearning, so there is no point in saying that this makes unpruning special; (2) the fact that networks cannot change topology is imposed by the authors (pruning algorithms could also adopt the unlearning objective in the mask search process), and does not necessarily mean is a bad thing as per my understanding; (3) generally speaking, is also true for normal unlearning, and conflates a legal requirement/desiderata with a technical difficulty; (4) I fail in understanding how the fact that pruning learns from data makes un-pruning non-trivial.
- Also, I believe that there are multiple contributions compressed in a single paper, which makes the reading often unclear and heavy.

**Methodological and technical limitations:**
- The algorithm relies on lowest weight magnitude pruning: this is a very simple heuristic which was only relevant 10 years ago, and makes the overall contribution, from the pruning perspective, a bit sloppy. In this regard, instead, I suggest the authors to consider embedding the unlearning objective into the mask-search process. There are in fact multiple ways these days to optimize the mask through some kind of approximation, and avoid using a very simple heuristic such as magnitude-based ones (consider for instance the paper you mentioned by Sehwag et al.).
- The authors fail in specifying that the correctness of the proposed un-pruning algorithm is ensured as long as the original network was pruned using the same magnitude-based heuristic. If that is not the case, the original and unpruning method used to sparsify the network would be misaligned.
- The problem setup mentions that pruned parameters are frozen, but this is not properly motivated. While I understand that the chosen architecture is typically data-dependent, one could in theory still re-optimize the remaining parameters. It would be interesting to see whether retraining with the same mask achieves similar results, rather than adopting the proposed approach and re-building the sparse topology.

**Writing and presentation**:
- While reading, the theoretical contribution is a bit out of the blue and is mostly a listing of formulas. The authors should consider organize this into theorems/propositions/lemmas/definitions when possible, and guide the readers through them.
- The algorithm’s property that it finds a new mask without retraining and repruning is not introduced until the algorithm section, making the setup confusing.
- I am not sure that this is the best solution to present the two contributions. The MIA-related contribution would probably benefit from a slightly separate narrative/prioritization.
- Figure 1 is overly-simplistic. If it was intended to illustrate the difficulty of un-pruning, it should be replaced with a more technical/informative one.  Also, some other minor problems are: (i) figures should have increased caption fontsize; (ii) there are multiple inconsistent and wrong plural forms; (iii) citations could benefit from being input with either citep or citet.

**Questions:**

I will write the questions by sticking to the suggested weaknesses:
1. Could you clarify why you chose the term un-pruning?
2. Why are the terms retraining and repruning preferred over training and pruning? Do they indicate a specific methodological distinction?
3. In the introduction, you claim that un-pruning is non-trivial. Could you provide a stronger justification for this statement beyond the four listed points?
4. Why did you decide to freeze pruned parameters instead of re-optimizing the remaining ones? Did you compare against more simply retraining with the same mask?
5. The proposed algorithm assumes lowest-weight-magnitude (LWM) pruning. How would it perform with alternative pruning strategies or learned masks?
6. Is the correctness of the proposed method guaranteed only when the same pruning heuristic (LWM) is used both before and after unlearning?
7. The theoretical analysis appears somewhat disconnected from the main algorithm. Could you clarify its purpose and practical implications?
8. The MIA-related analysis and the introduction of a new metric feel somewhat separate from the un-pruning contribution. Could you elaborate on how these parts relate to the central research question?

Curiosities:
- Could the unlearning objective be directly embedded into the mask optimization process, instead of relying on simple magnitude-based pruning?

---

### Official Review · Reviewer_AfD7 · 2025-10-28

**Soundness:** 3
**Presentation:** 3
**Contribution:** 2
**Rating:** 4
**Confidence:** 3

**Summary:**

The manuscript introduces the concept of un-pruning, arguing that the 'right to be forgotten' in machine learning should not only apply to model weights but also to the sparse topologies induced by pruning. It investigates the relationship between pruning and data dependency, and proposes algorithms to remove the influence of data on sparse models, exploring how forgetting can be realized in such sparse architectures. However, there remain several issues that merit further attention:

**Strengths:**

1. The manuscript proposes a new research question: how to eliminate the impact of deleted data on sparse model pruning topology.
2. The manuscript proposes an "un-pruning" algorithm that can approximate pruning topologies driven by retained data without the need for expensive re-training and re-pruning. The algorithm can integrate with any existing unloading algorithm and is suitable for both unstructured and structured topologies.
3. The manuscript points the vulnerability of MIA as an unloading evaluation indicator, and designs a new performance indicator to evaluate the quality of sparse model unloading.

**Weaknesses:**

1. The un-pruning strategy is mainly suitable for unstructured pruning algorithms that do not introduce additional learnable parameters for pruning. It is not used for structured pruning methods that use extra parameters to learn the pruning structure.
2. The legends are too small, which affects readability, for example in Fig 6.
3. The proposed Un-Pruning algorithm densifies the model, then applies the forgetting algorithm on this dense model, and finally prunes the weights. The authors do not provide a convincing justification for why this densification can effectively simulate training from scratch followed by pruning.

**Questions:**

1. If standard machine forgetting algorithms already make models behave like they were retrained on the remaining data, why modify the model topology additionally? Doesn't behavioral similarity mean "forgetting" that meets regulatory requirements?
2. If the forgetting algorithm itself induces structural changes (for example, pruning-based unlearning), it remains unclear whether the proposed Un-Pruning framework would still provide improvements or whether its effect would be neutralized. It is recommended that the authors clarify whether this framework is beneficial across different types of forgetting algorithms, such as gradient-based, distillation-based, or pruning-based unlearning methods.
3. Algorithm 1 relies on several heuristic steps. For example, in each iteration, it 'unfreezes' and reinitializes parameters, and then 'grows' the topology by adding p_θ% of non-zero parameters. What is the rationale behind the choice of these hyperparameters? These steps appear to be guided by intuition rather than a rigorous theoretical derivation.
4. In Chapter 5, the manuscript slightly modifies the “training/testing ratio” in shadow dataset to demonstrate that the MIA metric can be manipulated. However, the authors do not specify the standard experimental protocol for MIA, making the conclusion unconvincing.
5. Experiments have shown that "original initialization" is better than "random initialization". Why does restoring to the initial weights before model training began help better approximate pruning topologies driven by retained data?

---

### Official Review · Reviewer_aapr · 2025-10-30

**Soundness:** 2
**Presentation:** 1
**Contribution:** 2
**Rating:** 2
**Confidence:** 4

**Summary:**

The paper explores the problem of machine unlearning in sparse (pruned) models. While traditional unlearning methods focus on removing the influence of deleted data from fully dense networks, this work observes that deleted data can also affect the pruned topology, that is, the structure of sparsity resulting from pruning.
To address this, the authors introduce the new concept of “un-pruning” which aims to eliminate the residual influence of forgotten data on a model’s pruning mask. They propose an un-pruning algorithm that reconstructs a pruned topology using only the retained data, ensuring that the final sparse model reflects the updated training set.
The proposed un-pruning algorithm integrates any standard unlearning algorithm as a subroutine within the un-pruning process, where unlearning is performed on a temporarily dense version of the model before reapplying sparsity constraints.
For empirical validation, the authors conduct experiments on three datasets: CIFAR-10, FashionMNIST, and ImageNet, using three architectures: ResNet-18, AlexNet, and ViT. The results compare how the un-pruning strategy combined with various existing unlearning methods  performs against the gold baseline of retraining and re-pruning from stracth on the retain set.

**Strengths:**

- The paper introduces a problem that, at least to me, appears novel and unexplored: unlearning in sparse models.
- The paper integrates a suitable range of existing unlearning algorithms within the proposed un-pruning framework, which helps demonstrate the generality of the approach.

**Weaknesses:**

- The presentation quality could be significantly improved, as the current form makes the paper challenging to read and interpret. Additional comments are provided below.
- The section: Analysis of Un-Pruning is very difficult to follow due to numerous typos or errors (I listed some of them below) and unclear explanations. The discussion of the differentiable pruning strategy is confusing, as the actual algorithm relies on hard, discrete mask updates based on magnitude thresholding. The proposed Gumbel-Softmax formulation appears conceptually disconnected from the described implementation.
- In all the plots with “Unlearning Data Accuracy” on why axes. “Unlearning Data Accuracy” is not defined in the paper, I assume it is the accuracy on the forget set (the "deleted data") if this is the case, retraining from scratch should have the lowest UA. Here, retraining+repruning has almost the same UA as the original model, i.e. not much lower.
- The code is not avalible at least to me when I went to visit the code for every file I had the message "The requested file is not found."

Minor:

- Ambiguous Mask Update
Line 4 says "set $\Theta_p$ mask indexes whose corresponding parameters have the highest magnitude values as 1" but doesn't clarify whether this is   $\Theta_p$ of all parameters or  $\Theta_p$  of currently pruned parameters. The intended meaning is the latter, but this should be explicit.
- The term “deleted data” is used implicitly throughout the paper to denote training samples designated for removal or forgetting, though it is never explicitly defined.



Typos:
-  in Algorithm 1, line 178 of the PDF there are two consecutives commas “, ,”
- line 205: “unstructured turning algorithms “-> should be "pruning "
- line 257  there’s an inconsistency in notation. The authors are describing p as the sampling probability, but in the equation they use P($\Theta$)  as the argument inside the log.
-  missing space in line 86, 352
- Eq 17 diag(0), I assume this is a typo.
- Line 282 and 283 introducing the symbol $\sigma$ without any reason since it is not used after.
-  the symbol $\Theta_0$ is used in the algorihtm as initialized parameters and also as component of the modle (line 247).

Inconsistencies/errors:
-  line 257 is diffult to understand:   as defined by the authors p is the probability of sampling parameters from some random learning algorithm. What does it mean some random learning algorihtm? shoud it be an unlearnign algorithm? how is this probability defined?
- formula 15 is incorrect. it collapse to 0!  The orignal paper cited by the authros (Thud et al.), actually has a different form for approximating the oerall changes, they also have. The critical difference is that the Hessian is applied to the sum of all previous gradients plus all previous correction terms, not just the previous correction term alone. The incorrect simplified version misses the gradient accumulation entirely, making it mathematically meaningless.
- Looking at the definitions:  equation 11 is the hessian  is computed with respect to parameters. Equation 13:  $\lambda(M) := \max{\lambda_j(H),1 }. Why should this depend on M? The mask M doesn't appear anywhere in how the Hessian is computed.
-  Equation 16 is attempting to show how the parameter update changes when you introduce the mask, it’s not clear where the external Where does M^{t-1} comes from (the one after the \eta) . I suspect they're trying to say: during un-pruning, parameter updates only affect masked (non-pruned) parameters but the way it's written in Equation 16 doesn't clearly express this, and the different masks  it's not clear to me how they are applied.
- The analysis tries to connect un-pruning to unlearning via unrolled SGD, but never actually anlayzes what the algorithm does: iteratively adding parameters back by flipping mask bits from 0 to 1, re-initializing them, performing unlearning on the denser model, then finally repruning.

**Questions:**

- How does the choice of $ \p_\theta$ influences the perfromances of the algorithm?
- Can you provide more details on the type of inizilaization chosed for $\Theta_0$.
See also Weaknesses

---

### Official Review · Reviewer_eqEE · 2025-11-01

**Soundness:** 2
**Presentation:** 1
**Contribution:** 3
**Rating:** 2
**Confidence:** 4

**Summary:**

This paper proposes topology unlearning to remove data influence by un-pruning, which iteratively update the pruning mask. They discover that pruning mask is data-dependent and should be included in unlearning, and the proposed method serves as a plug-in-and-play module to improve existing unlearning algorithms.

**Strengths:**

1. The discovery of the pruning mask's data dependency is meaningful, and it inspires future work. The intuition and theoretical analysis of updating pruning mask is sound.

2. The proposed method is plug-in-and-play (or can be seen as a regularization), so it can be used to enhance existing unlearning methods.

3. This paper challenges MIA with plausible reasoning, and proposes their alternative. I am somewhat convinced by Section 5, though I think we can still refer to MIA with some aid metrics.

**Weaknesses:**

1. I am confused by the retrained+repruned goal, why is it the "gold standard"? In conventional unlearning literature, including sparsity-based unlearning work, e.g., [1], the gold standard is mostly the retrained model. Previous sparsity-inspired methods adopt sparsification to approach the retrained model, e.g. zeroing out parameters that can be activated by forget samples, or in other words parameters not important to retain samples and model performance. If the method only targets pruned, retrained model for deployment purpose, then it is quite limited too.

2. The experiment section seems to lack baseline results for all unlearning methods for all tables and figures, e.g. we don't have SCRUB vs. un-pruning(SCRUB)? Then we cannot evaluate the effectiveness of the proposed add-on method.

3. No ablation study to remove component(s) of un-pruning etc.

4. The authors should also compare with more recent unlearning algorithms with fixed pruning, such as [1] and [2], because the proposed method can be considered as a direct upgrade that unfreezes the masking.

5. The authors claim to experiment with various networks and datasets, but they are poorly organized, e.g. some plots/tables only present some of the claimed network+dataset settings. The figure/table captions do not always include the settings, and the paper does not specify in the experiment sections either.

6. It makes sense that restoring to original weights performs better than re-initializing with small non-zeros as there can be various discrepancies such as weight scales. But how much overhead does storing original weights cause? And since pruned model + saved weights that are pruned = original number of parameters, how much does it actually save from sparsification, just less gradient bookkeeping? The paper also lacks quantitative results regarding this.

[1] Jia, Jinghan, et al. "Model sparsity can simplify machine unlearning." Advances in Neural Information Processing Systems 36 (2023): 51584-51605.

[2] Fan, Chongyu, et al. "Salun: Empowering machine unlearning via gradient-based weight saliency in both image classification and generation." arXiv preprint arXiv:2310.12508 (2023).

**Questions:**

Please see Weaknesses for details. In summary, I find the analysis that pruning mask is data-dependent is interesting, and the resulting mask update method (unpruning) sounds plausible. However, the presentation of the paper in experiment settings is poor, lacking various kinds of experiment settings for fair comparisons and evaluation, and the remaining presentation is also not organized well. Moreover, the unlearning goal is not conventional and needs further clarification. Based on the motivation and strengths, I think this work has good potential and provides new insights (so I score 3 for contribution), but the current presentation is not good enough to convince me for the method's effectiveness and thus recommending acceptance.

---

### Note · Authors · 2025-12-02

I have read and agree with the venue's withdrawal policy on behalf of myself and my co-authors.